# Effects of Transient Receptor Potential Cation 5 (TRPC5) Inhibitor, NU6027, on Hippocampal Neuronal Death after Traumatic Brain Injury

**DOI:** 10.3390/ijms21218256

**Published:** 2020-11-04

**Authors:** Min Kyu Park, Bo Young Choi, A Ra Kho, Song Hee Lee, Dae Ki Hong, Jeong Hyun Jeong, Dong Hyeon Kang, Beom Seok Kang, Sang Won Suh

**Affiliations:** 1Department of Physiology, College of Medicine, Hallym University, Chuncheon 24252, Korea; bagmingyu50@gmail.com (M.K.P.); bychoi@hallym.ac.kr (B.Y.C.); rnlduadkfk136@hallym.ac.kr (A.R.K.); sshlee@hallym.ac.kr (S.H.L.); zxnm01220@gmail.com (D.K.H.); jd1422@hanmail.net (J.H.J.); ttiger1993@gmail.com (B.S.K.); 2Department of Medical science, College of Medicine, Hallym University, Chuncheon 24252, Kangwon-Do, Korea; ehdgus6312@gmail.com

**Keywords:** traumatic brain injury, zinc, NU6027, transient receptor potential cation channel 5, neuronal death

## Abstract

Traumatic brain injury (TBI) can cause physical, cognitive, social, and behavioral changes that can lead to permanent disability or death. After primary brain injury, translocated free zinc can accumulate in neurons and lead to secondary events such as oxidative stress, inflammation, edema, swelling, and cognitive impairment. Under pathological conditions, such as ischemia and TBI, excessive zinc release, and accumulation occurs in neurons. Based on previous research, it hypothesized that calcium as well as zinc would be influx into the TRPC5 channel. Therefore, we hypothesized that the suppression of TRPC5 would prevent neuronal cell death by reducing the influx of zinc and calcium. To test our hypothesis, we used a TBI animal model. After the TBI, we immediately injected NU6027 (1 mg/kg, intraperitoneal), TRPC5 inhibitor, and then sacrificed animals 24 h later. We conducted Fluoro-Jade B (FJB) staining to confirm the presence of degenerating neurons in the hippocampal cornus ammonis 3 (CA3). After the TBI, the degenerating neuronal cell count was decreased in the NU6027-treated group compared with the vehicle-treated group. Our findings suggest that the suppression of TRPC5 can open a new therapeutic window for a reduction of the neuronal death that may occur after TBI.

## 1. Introduction

A traumatic brain injury (TBI) can affect people of all ages, races, and genders [1]. TBI refers to brain damage caused by external forces, usually caused by sudden blows to the head, most commonly caused by traffic accidents, violence, or falls [1]. TBI induces brain contusions, lacerations, and focal or diffuse intracranial hemorrhages, ultimately leading to neuronal death [2]. Primary external injury is directly related to the immediate physical trauma to brain structures. Secondary injuries, however, can occur within minutes or some days following the initial impact and are driven by altered molecular, chemical, and inflammatory cascades [3]. TBI-induced neuronal death is an active and energy-dependent process resulting from a continuous maladaptive signaling loop that is initiated following the accumulation of glutamate and zinc [4]. In addition, TBI-induced reactive oxygen species (ROS) production through nicotinamide adenine dinucleotide phosphate (NADPH) oxidase activation triggers lipid peroxidation and protein degradation [5,6,7].

Zinc is the most abundant metal ion in the brain and is an essential element for the growth and division of all cells and the development of tissues [8]. Zinc has a positive effect on bone metabolism, imitating growth hormone (GH), or insulin-like growth factor 1 (IGF-1) [9]. Zinc is important for the sequence-specific DNA binding by various proteins that regulate transcription and DNA synthesis. Zinc is also loosely bound to proteins, acting as a component of the catalytic region, and controlling the morphological and structural capacity of the enzyme [10,11]. Furthermore, zinc modulates cellular differentiation and normalizes several physiological functions [10]. Zinc can be released from presynaptic terminals after epilepsy, cerebral ischemia, and TBI [12,13,14]. Following this, zinc is translocated and accumulates in the postsynaptic neurons; this is an important driver of excitotoxic neuronal injury after injuries [12,15,16]. Accumulation of excessive levels of zinc in the postsynaptic neuron activates NADPH oxidase, which produces ROS and eventually induces oxidative stress [17,18]. During oxidative stress, the zinc-binding protein metallothionein-3 releases free zinc into the cytoplasm, which facilitates neuronal injury [19].

Transient receptor potential (TRP) channels are located throughout the body and have a non-selective cation permeability. There are seven subtypes of TRP channels: transient receptor potential cation, transient receptor potential melastatin, transient receptor potential vanilloid, transient receptor potential ankyrin, transient receptor potential polycystic, Transient receptor potential mucolipin, and TRPN (the name no mechanoreceptor potential C) [20]. Specifically, TRPC5 is expressed both in the peripheral sensory nerves and in the central nervous system [21]. TRPCs are located in the plasma membrane and regulate the influx of calcium and sodium ions [22]. Transient receptor potential (TRP) channels, TRPC4 and TRPC5 are homogeneous proteins distributed in various regions of the brain, especially the hippocampus [23]. The TRPC5 channel contributes to the foundation of spatial working memory and the mechanism that regulates the flexible relearning by promoting the proper synaptic transmission of hippocampal neurons [24]. However, excessive activation of the TRPC5 channel under pathological conditions continues to increase the transport of cytotoxic Ca^2+^, activating calmodulin-dependent protein kinase, and calpain-caspase, which ultimately leads to neuronal death [14,25]. It is reported that the TRPC5 channel is similarly activated after epilepsy, resulting in an increase in neuronal death in the hippocampus [14,26]. TRPC5 KO mice showed decreased excitability during epilepsy and reduced normal spatial learning, and also showed a reduction of neuronal death [14,27].

In a previous study, cyclin-dependent kinase (CDK) inhibitor NU6027 was shown to have a potential protective effect on hydrogen peroxide-induced calcium influx, oxidative damage, and subsequent cortical neuronal death. It was asserted that NU6027 blocks seizure-induced hippocampal neuronal death via the inhibition of TRPC5 [14,27]. However, the efficacy of NU6027 on TBI-induced neuronal death is not yet known.

In the present study, we evaluated our hypothesis that TBI-induced neuronal death is mediated by TRPC5 activation and that administration of TRPC5 inhibitor can decrease TBI-induced zinc accumulation, oxidative stress, and, ultimately, neuronal death. To test our hypothesis, we used a controlled cortical impact injury (CCI) model for TBI using adult rats.

## 2. Results

### 2.1. TRPC5 Inhibitor Reduces TBI-Induced Hippocampal Neuronal Death

In this experiment, male rats were used to minimize the error in the experiment by eliminating the effects of hormones from the female rat menstrual cycle. The NU6027-treated group demonstrated a dramatically reduced number of degenerating hippocampal neurons. To confirm the presence of neuronal death, we detected degenerating neurons by Fluoro-Jade B staining in the hippocampal cornus ammonis 3 (CA3) region 24 h after TBI. (Figure 1A). Comparing the TBI-vehicle group and the TBI-NU6027 (1 mg/kg) treated group, the number of degenerating neurons was dramatically higher in the TBI-vehicle group than the TBI-NU6027 (1 mg/kg) treated group. Figure 1B displays the quantified FJB (+) neurons in the CA3 region. The NU6027 treated group displayed a reduction of FJB (+) neurons by around 66% in the CA3 (TBI-vehicle, 139.8 ± 31.1; TBI-NU6027, 47.4 ± 17.3) region compared to the vehicle-treated group. * *p* < 0.05 vs. vehicle-treated group.

### 2.2. TRPC5 Inhibitor Reduces TBI-Induced Zinc Accumulation

The NU6027 treated group displayed a dramatically reduced number of zinc-specific TSQ (+) neurons. To confirm zinc accumulation in hippocampal neurons, we detected zinc-specific TSQ (+) neurons by N-(6-methoxy-8-quinolyl)-para-toluene sulfonamide (TSQ) from the hippocampal CA3 region 12 h after TBI. (Figure 1C). The number of zinc-specific TSQ (+) neurons was higher in the TBI-vehicle group than the TBI-NU6027 (1 mg/kg) treated group. Figure 1D displays the quantified TSQ (+) neurons in the CA3 region. The NU6027 treated group displayed a greater reduction of TSQ(+) neurons (about 56%) in the CA3 (TBI-vehicle, 41.2 ± 5.6; TBI-NU6027, 17.9 ± 3.9) region compared to the TBI-vehicle group. * *p* < 0.05 vs. vehicle-treated group.

### 2.3. TRPC5 Inhibitor Decreases TBI-Induced Hippocampal Dendrite Loss

The NU6027 treatment group displayed a significant reduction in microtubule loss. To confirm microtubule loss, we detected microtubule-associated protein 2 (MAP2) from the hippocampal cornus ammonis 3 (CA3) region 24 h after TBI, and this differed significantly between the treatment vs. the non-treatment groups (Figure 2A). The sham-vehicle and sham-NU6027 groups displayed no difference in MAP2 fluorescence signals. On the other hand, we found that the group treated with NU6027 (1 mg/kg) showed a decreased microtubule intensity in the hippocampal CA3 region compared to the TBI-vehicle group. Figure 2B displays the MAP2 intensity in the CA3 region. The NU6027 treatment group exhibited an increased MAP2 intensity by about 61% in the CA3 (TBI-vehicle, 11.3 ± 4.9; TBI-NU6027, 29.5 ± 12.9) region compared with the vehicle-treated group. * *p* < 0.05 vs. vehicle-treated group.

### 2.4. TRPC5 Inhibitor Decreased TBI-Induced Oxidative Injury

The NU6027 treatment group displayed a dramatically reduced intensity of oxidative stress. To confirm the level of oxidative stress, we performed 4-hydroxynonenal (4HNE) from the hippocampal CA3 region 24 h after TBI, and this differed significantly among groups (Figure 3A). The sham-vehicle and sham-NU6027 groups displayed no difference in 4HNE fluorescence signals. On the other hand, we found that the group treated with NU6027 (1 mg/kg) showed a decreased oxidative stress in the hippocampal CA3 region compared to the TBI-vehicle group. Figure 3B displays the 4HNE intensity in the CA3 region. The NU6027 treatment group displayed a 71% reduction of 4HNE intensity in the CA3 (TBI-vehicle, 42.9 ± 21.4; TBI-NU6027, 12.1 ± 6.6) region compared with the vehicle-treated group. * *p* < 0.05 vs. vehicle-treated group.

### 2.5. TRPC5 Inhibitor Reduces TBI-Induced Hippocampal TRPC5 Channel Expression

The NU6027 treatment group displayed a dramatically reduced TRPC5 channel activation. To confirm TRPC5 channel activation, we performed TRPC5 staining in the hippocampal CA3 region 24 h after TBI. (Figure 4A). The sham-vehicle and sham-NU6027 groups displayed no difference in TRPC5 fluorescence signals. On the other hand, we found that the group treated with NU6027 (1 mg/kg) showed a significant decrease in TRPC5 channel activation in the hippocampal CA3 region compared to the TBI-vehicle group Figure 4B displays the TRPC5 intensity in the CA3 region. The NU6027-treated group displayed a 55% reduction of TRPC5 intensity in the CA3 (TBI-vehicle, 36.9 ± 6.1; TBI-NU6027, 16.5 ± 3.3) region compared with the TBI-vehicle group. * *p* < 0.05 vs. vehicle-treated group.

### 2.6. TRPC5 Inhibitor Reduces TBI-Induced Hippocampal Glial Activation

The NU6027 treatment group displayed significantly reduced glial activation compared to the other groups. To detect glial activation, we applied Glial fibrillary acidic protein (GFAP) and ionized calcium-binding adapter molecule 1 (Iba-1) in the hippocampal CA3 region 24 h after TBI (Figure 5A,C). The sham-vehicle and sham-NU6027 groups displayed no difference in fluorescence signal intensity. On the other hand, we found that the group treated with NU6027 (1 mg/kg) showed a decreased glial activation in the hippocampal CA3 region compared to the TBI-vehicle group. Figure 5B,D displays the GFAP intensity and Iba-1 intensity in the CA3 region. The NU6027 treatment group displayed a 41% reduction of GFAP intensity in the CA3 (TBI-vehicle, 29.9 ± 5.4; TBI-NU6027, 17.6 ± 3.4) region compared with the TBI-vehicle group. Moreover, the NU6027 treatment group displayed a 25% reduction of Iba-1 intensity in the CA3 (TBI-vehicle, 26.7 ± 5.0; TBI-NU6027, 19.8 ± 4.6) region compared with the TBI-vehicle group. * *p* < 0.05 vs. vehicle-treated group.

### 2.7. TRPC5 Inhibitor Reduces TBI-Induced Hippocampal Neuron Loss

NU6027 treatment groups exhibited a significant increase in surviving neurons compared with the other groups. To quantify the live neurons, we applied neuronal nuclei (NeuN) in the hippocampal CA3 region 1 week after TBI (Figure 6A). The sham-vehicle and sham-NU6027 groups displayed no difference in the number of live neurons but we found that the group treated with NU6027 (1 mg/kg) showed an increased number of live neurons in the hippocampal CA3 region compared to the TBI-vehicle group. Figure 6B displays the live neuron count in the CA3 region. The NU6027-treated group increased the number of live neurons by about 46% in the CA3 (TBI-vehicle, 24 ± 171.3; TBI-NU6027, 461.9 ± 82.5) region compared to the TBI-vehicle group. * *p* < 0.05 vs. vehicle-treated group.

### 2.8. TRPC5 Inhibitor Reduces TBI-Induced Cognitive Impairment

The NU6027 treatment group displayed a significant reduction in symptoms that mimic those seen in cognitive disorders. To evaluate the neurological function, we carried out a modified neurological severity score (mNSS) test 1 week after TBI (Figure 7). The sham-vehicle and sham-NU6027 groups displayed no difference in mNSS score. However, the NU6027 treatment group had a reduced mNSS score compared to the TBI-vehicle group. * *p* < 0.05 vs. vehicle-treated group.

## 3. Discussion

In the present study, we tested whether NU6027 inhibits TBI-induced neuronal death via inhibition of transient receptor potential cation channel 5 (TRPC5) channels. TRPC channels are subdivided into seven isotypes (TRPC1–TRPC7) and, among them, TRPC5 are abundantly expressed in the rat brain [23,25,28]. Several studies have demonstrated that TRPC5 over-activation is involved in seizure-induced neuronal death [14,25,27]. In the present study, in one cohort of adult male rats, we present evidence that CCI-induced increases in neuronal death, zinc accumulation, dendritic loss, oxidative damage, TRPC5 expression, and gliosis 24 h after injury are all reduced by post-impact injection of NU6027. In a second cohort, we show that motor function, as assessed by modified neurological severity score (mNSS) 1 week after CCI, was improved in NU6027 treated rats. Therefore, NU6027 may be a potential therapeutic agent for preventing TBI-induced neuronal death.

The mechanisms for TBI-induced brain damage are still not fully understood. However, we suggest a new means for preventing TBI-induced injury, that is, oxidative stress-related cell death cascades. When TBI occurs via a severe external force, the primary problem is a brain edema, which causes increased intracranial pressure and directly contributes to higher mortality observed in severe TBI-suffered patients [29,30]. TBI-induced primary injury leads to secondary injury, which is primarily via ROS production through NADPH oxidase activation [7], glial activation [17], loss of microtubules [31], and the accumulation of zinc in post-synaptic neurons from the excessive release of vesicular zinc [32,33]. Recently, several studies asserted that a moderate zinc concentration regulates physiological functions, but that dysregulation of the ionic gradient is devastating to the central nervous system [34,35]. Besides, oxidative stress has been known as a key aggravator in the acute brain disease field [36]. Consequently, we recognized that the regulation of zinc and ROS-mediated oxidative stress was important to alleviate TBI-induced neuronal damage [33]. We investigated whether the application of the cyclin-dependent kinase inhibitor NU6027 exhibits a neuroprotective effect through the inhibition of TRPC5 and ROS production after a TBI [14]. (Figure 8).

When a TBI occurs, a huge amount of vesicular zinc is released from the terminal before the synapse, which is then translocated into neurons through the postsynaptic TRPC5 channel [14,15]. Following this, vesicular zinc is introduced into neurons and protein kinase c (PKC) is activated by vesicular zinc, and subunits P47 (phox) and P67 (phox) of NADPH are activated by protein kinase c (PKC) [39,40,41,45]. The P47 (phox) and p67 (phox) of the NADHPH subunits activated by PKC stimulate the neuronal membrane to activate gp91 (phox) and p21 (phox), both being present in the membrane, thereby generating superoxide production and eventually leading to increased oxidative stress [39,40,41,46]. It also activates polymerase-1 (PARP-1) in a damaged state, resulting in severe cellular damage to neurons, as it requires ATP generation and quickly consumes available supplies [39,40,41,46]. Ultimately, activated NADPH oxidase increases the overall oxidative stress levels [15,45]. When oxidative stress occurs, free zinc is liberated from metallothionein, a zinc-binding protein, and this free zinc again further increases oxidative stress [14,19,34,42,43].

Several TRP channels are regulated by metal ions, of which TRPA1 and TRPC5 channels are regulated by zinc in metal ions [14,47,48]. The oxidative stress produced by a TBI leads to the accumulation of large concentrations of free zinc in neurons [49]. Free zinc accumulated in neurons activates the TRPC5 channel, and this activated TRPC5 channel may increase the influx of toxic levels of calcium or zinc ions [14,44]. The administration of NU6027 after TBI reduces oxidative stress, which reduces zinc accumulation [14]. Originally, NU6027 was an inhibitor for ATR and cyclin-dependent kinase1 and 2, which was mainly used to inhibit the division of cancer cells through the suppression of cell growth [50,51]. However, in previous studies it has been shown that NU6027 inhibits TRPC5 activity in channels present in cortical neurons that are activated during epilepsy, reducing calcium influx [14,52]. NU6027 also reduced H_2_O_2_ increase in cortical neurons during epilepsy, indicating that NU6027 has reduced the activity of TRPC5 channels, ultimately reducing neuronal death. [14]. Therefore, our study demonstrated that following a TBI the activation of TRPC5 channels can be reduced through the administration of NU6027, thus reducing the level of oxidative stress and reducing the accumulation of free zinc, ultimately leading to enhanced neuronal survival.

The reason why we chose the hippocampal CA3 region in this study is that our CCI TBI model directly damages the hippocampal CA3 region, which is known to have a high level of TRPC1 and TRPC5 channel expression [22,53,54]. In this experiment, a male rat was used to minimize the error in the experiment by eliminating the effects of hormones from the female rat menstrual cycle.

In the present study, we found that the number of degenerating neurons in the hippocampal CA3 regions was significantly reduced in the group treated with NU6027. The number of neurons with zinc accumulation in the hippocampal CA3 region was also significantly reduced in the NU6027 treatment group compared to the TBI vehicle group [14]. These results show that NU6027 reduces the accumulation of vesicular zinc in neurons through the TRPC5 channel, which contributed to the reduction of neuronal death after TBI. We confirmed that when TBI occurs, free zinc accumulates in the neuron due to oxidative stress [19,42,43], which activates the TRPC5 channel, causing excessive cycling of free zinc and calcium entry, eventually leading to neuronal death [14].

TBI damage leads to microtubule loss. We confirmed microtubule loss through MAP2 staining. The results showed a significant reduction in microtubule loss in the NU6027 treatment group in the hippocampal CA3 regions, where the microtubule density was confirmed through MAP2 staining. TBI damage also leads to oxidative stress. We confirmed the presence of oxidative stress through 4HNE staining. The results showed a significant reduction in oxidative stress via monitoring of the 4HNE fluorescence signal intensity in the NU6027 treatment group in the hippocampal CA3 region. This is because free zinc accumulates, damaging the microtubules, and promoting oxidative stress [33,55]. NU6027 blocks zinc from entering through the TRPC5 channel, thereby reducing the occurrence of zinc-induced microtubule loss and oxidative stress.

The TRPC5 channel is inappropriately activated because of TBI damage. We confirmed TRPC5 channel expression through immuno-staining. The NU6027 treatment group showed a significant reduction in TRPC5 channel expression in the hippocampal CA3 region. The TRPC5 channel is activated by free zinc produced by oxidative stress, which is inhibited by NU6027. Metallothionein, a zinc-binding protein present in neurons, releases a lot of free zinc when exposed to oxidative stress, activating the TRPC5 channel. NU6027 not only reduces H_2_O_2_, but also reduces oxidative stress and, as an inhibitor for the TRPC5 channel, it reduces the activity of the TRPC5 channel [14,19,42].

The microglia and astrocyte are activated by TBI injury [56,57]. NADPH oxidase is activated by TBI to generate ROS, which activates the microglia and astrocyte [58]. We confirmed microglia activation through Iba-1 staining. The NU6027 treatment group showed a significant reduction in microglial activation in the hippocampal CA3 region. We confirmed astrocyte activation through GFAP staining. Once again, the NU6027 treatment group showed a significant reduction in astrocyte activation in the hippocampal CA3 region. Oxidative damage was reduced by administering NU6027, reducing the activation of the microglia and astrocyte.

A week after the induction of TBI, we evaluated the number of live neurons through NeuN staining. The number of NeuN positive neurons in the hippocampal CA3 regions was increased in the NU6027 treatment group. Thus, we found that the administration of NU6027 improved the survival of hippocampal neurons a week after a TBI.

The cognitive impairment caused by TBI was evaluated using the modified neurologic severity scores (mNSS) test. Injury to the CA3 region in the hippocampus by a TBI can cause problems associated with the encoding, storage, and retrieval of memory [59,60]. The TBI-NU6027 treatment group performed better in the mNSS test than the TBI-vehicle group, confirming that the score of mNss was lower. We confirmed that NU6027 improved TBI-induced cognitive impairment.

In the present study, we have demonstrated that the suppression of TRPC5 activity by NU6027 reduces the accumulation of zinc, thereby reducing neuronal death. Therefore, our findings suggest that NU6027 can be a potential therapeutic tool to prevent TBI-induced neuronal death.

## 4. Materials and Methods

### 4.1. Ethics Statement

The present study was performed in accordance with the protocols of the Guidelines for the Use and Care of Laboratory Animals, allowed by the National Institutes of Health. Animal studies were conducted in accordance with the guidelines of the Committee on Animal Use for Study and Education at Hallym University (protocol no. Hallym-2019-68). We sacrificed mice under isoflurane anesthesia to minimize pain and suffering.

### 4.2. Experimental Animals

The present study used adult male Sprague-Dawley rats (SD-Rat) (age of 8 weeks, 300–350 g, DBL Co., Chungcheongbuk-do, Korea). In this experiment, male rats were used to minimize the error in the experiment by eliminating the effects of hormones from the female rat menstrual cycle. Rats were housed at three to four rats per cage under conditions of sustained temperature (22 ± 2 °C) and humidity (55 ± 5%). Animal room lights were managed automatically, turned on and off in a 12 h cycle (on at 6:00 a.m., off at 6:00 p.m.).

### 4.3. Controlled Cortical Impact Model for TBI

Male Sprague-Dawley rats (SD-Rat) were used as controls. Rats were deeply anesthetized with 1–1.5% isoflurane and a 25:75 mixture of oxygen/nitrous oxide (David Kopf Instruments, Tujunga, CA, USA). TBI was performed using an electromagnetic (EM) controlled cortical impact device (Impact One TM Stereotaxic Impactor, Richmond, IL, USA). Craniotomy was carried out using a portable drill and a 3.0 mm diameter hole was drilled over the right hemisphere (3.0 mm Lambda to the Bregma and 2.8 mm lateral to the midline). A velocity of 5 m/s and a strike depth of 3.0 mm flap tip-impactor was accolated down. All rats (300–350 g) were maintained at a core temperature of 36–37.5 °C during TBI surgery. Rats were sacrificed 24 h and 1 week after TBI.

### 4.4. NU6027 Administration

NU6027-treated groups were administrated NU6027 (1 mg/kg, i.p.) and 0.9% normal saline. NU6027 injected once immediately intraperitoneal after TBI-induced. NU6027(1 mg/kg, i.p.) was injected once per day for 1 week, after which animals were sacrificed. The experimental groups were divided into four groups: (1) sham-vehicle (saline only, n = 5); (2) sham-NU6027 (NU6027 only, n = 5); (3) TBI-vehicle (TBI + saline, n = 6); (4) TBI-NU6027 (TBI + NU6027, n = 6).

### 4.5. Brain Sample Preparation

TBI was induced in rats after 24 h and 1 week. They were deeply anesthetized using urethane (1.5 g/kg, i.p.) and then sacrificed. The animals were perfused transcranially with 0.9% saline, followed by 4% paraformaldehyde (PFA). Then, we harvested the brain tissue, which was fixed by 4% paraformaldehyde for 1 h. Fixed brains were moved in a 30% sucrose solution overnight for cryoprotection. After 2 days, the brains sank to the bottom of the 30% sucrose, and they were frozen for 10 min on the freezing medium. The brains were cut with cryostats at 30 μm thickness, and tissues were kept in a storage solution before being used for immunohistochemistry and immunofluorescence staining.

### 4.6. Confirmation of Hippocampal Neuron Degeneration

To confirm the degeneration of neurons after the traumatic brain injury (TBI), brain tissue was cut with cryostats at 30 μm thickness and was attached to gelatin-coated slides (Fisher Scientific, Pittsburgh, PA, USA). The slides were soaked in 100% ethanol for 3 min. After that, they were soaked in 70% ethanol for 1 min. The slides were then soaked in 0.06% potassium permanganate for 15 min in the shade. Slides were soaked in 0.001% FJB (Histo-Chem Inc., Jefferson, AR, USA) for 30 min after that, we washed brain tissue three times for 10 min in DW. Then, we mounted cover slides on the slides with DPX (Sigma-Aldrich Co., St. Louis, MO, USA). Brain tissue sections were observed through a fluorescence microscope (Olympus, Tokyo, Japan) via blue (450–490 nm) excitation light. FJB-positive cells were observed to locate the hippocampal CA3 region.

### 4.7. Confirmation of Hippocampal Zinc Accumulation

To confirm intraneuronal zinc accumulation, we performed TSQ staining after TBI. Rats were deeply anesthetized with 1–1.5% isoflurane and a 25:75 mixture of oxygen/nitrous oxide (David Kopf Instruments, Tujunga, CA, USA). Then, brain tissues were harvested quickly without perfusion. Brain tissues were cut with cryostats at 10 μm thickness in a −15 °C cryostat, and were then attached to gelatin-coated slides (Fisher Scientific, Pittsburgh, PA, USA) and dried. After that, slides were soaked in a solution for 1 min in a solution of 4.5 mmol/L TSQ (Enzo Life Science, Enzo Biochem, Inc., Farmingdale, New York, NY, USA, ENZ-52153), and were then washed for 1 min in normal 0.9% saline. Brain tissue sections were observed through a fluorescence microscope (Olympus, Tokyo, Japan) under 360 nm UV light and a 500 nm long-pass filter. Then, TSQ-positive neurons of the hippocampal region were counted by blind quantification.

### 4.8. Confirmation of Hippocampal Microtubule Loss

Microtubule loss was detected by microtubule-associated protein 2 (MAP2, green color) immunohistochemical staining. Brain tissues were stained with MAP2 antibodies (Alpha Diagnostic Intl. Inc., San Antonio, TX, USA). We cut the brain tissues and washed them three times for 10 min in 0.01 M PBS. Then, to block intracellular per-oxidase, we cut the brain tissues and they were soaked for 15 min at room temperature in 1.2% hydrogen peroxide. We then cut brain tissues and washed them three times for 10 min in 0.01 M PBS. Then, we cut the brain tissues and they were soaked in a PBS containing 0.3% TritonX-100 with polyclonal rabbit anti-MAP2 serum (diluted 1:200, Alpha Diagnostic Intl. Inc., San Antonio, TX, USA) overnight at 4 °C in an incubator. We then cut the brain tissues and washed them three times for 10 min in 0.01 M PBS. Then, we cut the brain tissues and they were immersed for 2 h at room temperature in a solution of Alexa-Fluor-488-conjugated donkey anti-rabbit IgG (diluted 1:250, Invitrogen, Grand Island, NY, USA) secondary antibody. Finally, we cut the brain tissues and washed them again, before cover slides were mounted with DPX (Sigma-Aldrich Co., St. Louis, MO, USA). The brain tissue was observed through a fluorescence microscope (Olympus, Japan). We analyzed the results with the Image J program to measure the microtubule damage and measured the mean gray value.

### 4.9. Confirmation of Hippocampal Oxidative Stress

Oxidative stress was detected by 4-hydroxyl-2-nonenal (4HNE, red color) immunohistochemical staining. The cut brain tissues were stained by 4HNE antibodies (Alpha Diagnostic Intl. Inc., San Antonio, TX, USA). The brain tissues were washed three times for 10 min in 0.01 M PBS. Then, to block intracellular peroxidase, we cut the brain tissues and they were soaked for 15 min at room temperature in 1.2% hydrogen peroxide. Brain tissues were then washed three times for 10 min in 0.01 M PBS. Then, brain tissues were soaked in a PBS containing 0.3% TritonX-100 with polyclonal mouse anti-4HNE serum (diluted 1:500, Alpha Diagnostic Intl. Inc., San Antonio, TX, USA) overnight in a 4 °C incubator. Brain tissues were then washed three times for 10 min in 0.01 M PBS and were then immersed for 2 h at room temperature in a solution of Alexa-Fluor-594-conjugated donkey anti-mouse IgG (diluted 1:250, Invitrogen, Grand Island, NY, USA) secondary antibody. Finally, the brain tissues were washed three times for 10 min in 0.01 M PBS cover slides were mounted on the slides with DPX (Sigma-Aldrich Co., St. Louis, MO, USA). The brain tissue was observed through a fluorescence microscope (Olympus, Japan), and we analyzed it using the Image J program to measure oxidative stress and measured the mean gray value.

### 4.10. Confirmation of Hippocampal TRCP5 Channel Expression Level

TRPC5 channel expression was detected by transient receptor potential channel 5 (TRPC5, red color) immunohistochemical staining. Brain tissues were stained by TRPC5 antibodies (Alomone Laboratories, Jerusalem, Israel). The brain tissues were washed three times for 10 min in 0.01 M PBS. To block intracellular peroxidase, the brain tissues were soaked for 15 min at room temperature in 1.2% hydrogen peroxide. Then, the brain tissues were washed three times for 10 min in 0.01 M PBS. After this, tissues were soaked in a PBS containing 0.3% TritonX-100 with polyclonal rabbit anti- TRPC5 serum (diluted 1:200, Alomone Laboratories, Jerusalem, Israel) overnight in an incubator at 4 °C. After this, the brain tissues were washed three times for 10 min in 0.01 M PBS and immersed for 2 h at room temperature in a solution of Alexa-Fluor-594-conjugated donkey anti-rabbit IgG (diluted 1:250, Invitrogen, Grand Island, NY, USA) secondary antibody. Finally, the brain tissues were washed three times for 10 min in 0.01 M PBS. Cover slides were mounted on the slides with DPX (Sigma-Aldrich Co., St. Louis, MO, USA) and the brain tissue was observed through a fluorescence microscope (Olympus, Japan). Using the Image J program, we measured the TRPC5 channel expression level and measured the mean gray value.

### 4.11. Confirmation of Hippocampal Glial Activation

To confirm glial activation, astroglia were detected by Glial fibrillary acidic protein (GFAP, green color) immunohistochemical staining, microglia were detected by ionized calcium-binding adapter molecule 1 (Iba-1, red color) immunohistochemical staining. For brain tissue staining we used polyclonal goat anti-GFAP serum (diluted 1:1000, Abcam, Cambridge, UK). Microglial activation was detected by ionized calcium-binding adapter molecule 1 (Iba-1, red color) immunohistochemical staining. For brain tissue staining we then used polyclonal rabbit anti-Iba-1 serum (diluted 1:500, Abcam). Sections were washed three times for 10 min in 0.01 M PBS and to block intracellular peroxidase, sections were then soaked for 15 min at room temperature in 1.2% hydrogen peroxide. Sections were then washed three times for 10 min in 0.01 M PBS. Sections were then soaked in a PBS containing 0.3% TritonX-100 with goat anti-GFAP serum (diluted 1:1000, Abcam) and rabbit anti-Iba-1 serum (diluted 1:500, Abcam) overnight in an incubator at 4 °C. Sections were again washed three times for 10 min in 0.01 M PBS and immersed for 2 h at room temperature in a solution of Alexa-Fluor-488-conjugated donkey anti-goat IgG (diluted 1:250, Invitrogen, Grand Island, NY, USA) secondary antibody and Alexa-Fluor-594-conjugated donkey anti-rabbit IgG (diluted 1:250, Invitrogen, Grand Island, NY, USA) secondary antibody. Finally, sections were washed three times for 10 min in 0.01 M PBS and covers were mounted on the slides with DPX (Sigma-Aldrich Co., St. Louis, MO, USA). We observed brain tissue sections using a fluorescence microscope (Olympus, Japan) and analyzed it using the Image J program to measure astrocyte and microglial activation and measure the mean gray value.

### 4.12. Confirmation of Hippocampal Live Neuron

Live neurons were detected by neuronal nuclei (NeuN) immunohistochemical staining. Brain tissue staining was carried out using NeuN antibodies (Billerica, Millipore Co., Billerica, MA, USA). Sections were washed three times for 10 min in 0.01 M PBS and then, to block intracellular peroxidase, sections were soaked for 15 min at room temperature in 1.2% hydrogen peroxide. Then, sections were washed three times for 10 min in 0.01 M PBS and were soaked in a PBS containing 0.3% TritonX-100 with polyclonal mouse anti-NeuN serum (diluted 1:500, Alpha Diagnostic Intl. Inc., San Antonio, TX, USA) overnight in a 4 °C incubator. Then, sections were washed three times for 10 min in 0.01 M PBS and were immersed for 2 h at room temperature in a solution of biotinylated anti-mouse IgG (diluted 1:250, Vector, Burlingame, CA, USA) secondary antibody. Then, sections were washed three times for 10 min in 0.01 M PBS and immersed for 2 h at room temperature in a solution of ABC compounds (Vector, Burlingame, CA, USA). Following this, sections were washed three times for 10 min in 0.01 M PBS. To visualize the immune response, sections were soaked in a 0.01 M PBS containing 0.06% 3,3′-diaminobenzidine (DAB ager, Sigma-Aldrich Co., St. Louis, MO, USA). Sections were washed three times for 10 min in 0.01 M PBS and cover slides were mounted on the slides with Canada balsam. We observed brain tissue sections using an Olympus IX70 inverted microscope (Olympus Co., Tokyo, Japan) and counted the live neurons of the hippocampal region, and analyzed the total number of NeuN-positive cells [61].

### 4.13. Confirmation of Behavior Test

To confirm that NU6027 treatment rescues TBI-induced neurological deficits, neurological function tests were conducted using a modified neurological severity score (mNSS) [62]. Tests were conducted at 1,2,3 and 7 days after TBI and sham control. The mNSS grade 18 means that all tasks failed, while grade 0 means that all tasks are performed successfully. Detailed descriptions of mNSS grade tests included (1) raising rats by tail (3 points), (2) placing rat on floor (3 points), (3) sensory tests (3 point), (4) beam balance tests (6 points), (5) reflex absence and abnormal movements (4 points) [63]. Rats were sacrificed a week after mNSS assessment.

### 4.14. Statistical Analysis

All the results of the present experimental cohorts were shown as the mean value ± standard error of mean (SEM). Comparisons between vehicle-treated and NU6027-treated rats were conducted using the non-parametric Mann–Whitney U test. Statistical significance was set at *p* < 0.05.

## 5. Conclusions

The present study suggests that NU6027 can potentially serve as a therapeutic tool for protection against neuronal death after TBI.

## Figures and Tables

**Figure 1 ijms-21-08256-f001:**
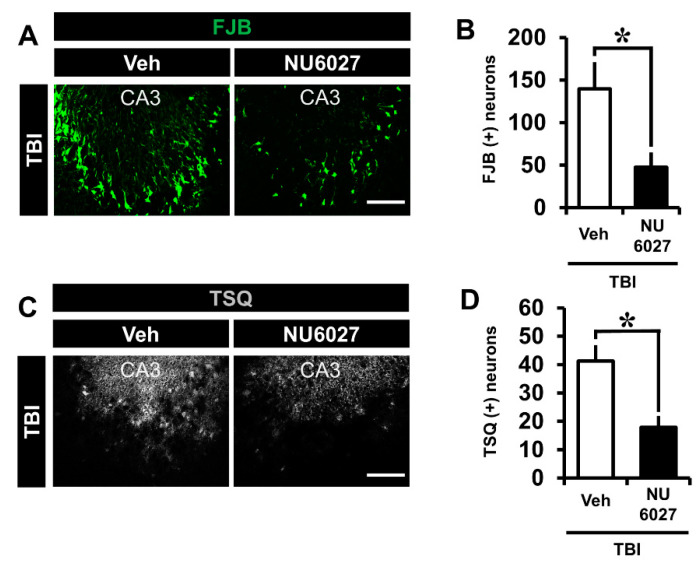
(**A**) Fluorescent images display degenerating neurons detected by Fluoro-Jade B (FJB, green color) from the hippocampal CA3 region 24 h after traumatic brain injury (TBI). NU6027 (1 mg/kg) post-treatment decreased the degenerating neurons in the hippocampal CA3 region when compared with the vehicle-treated group. (**B**) Bar graph indicates the number of degenerating neurons in the hippocampal CA3 region. (TBI-vehicle: *n* = 6; TBI-NU6027: *n* = 6). (**C**) Zinc-specific representative image displays TSQ (+) neurons detected by N-(6-methoxy-8-quinolyl)-para-toluene sulfonamide (TSQ) from the hippocampal CA3 region 12 h after TBI. (**D**) The bar graph indicates TSQ (+) neurons in the CA3 region (TBI-vehicle, *n* = 5; TBI-NU6027, *n* = 5). Data are mean ± S.E.M. * Significantly different from the vehicle-treated group, *p* < 0.05. Scale bar = 100 μm.

**Figure 2 ijms-21-08256-f002:**
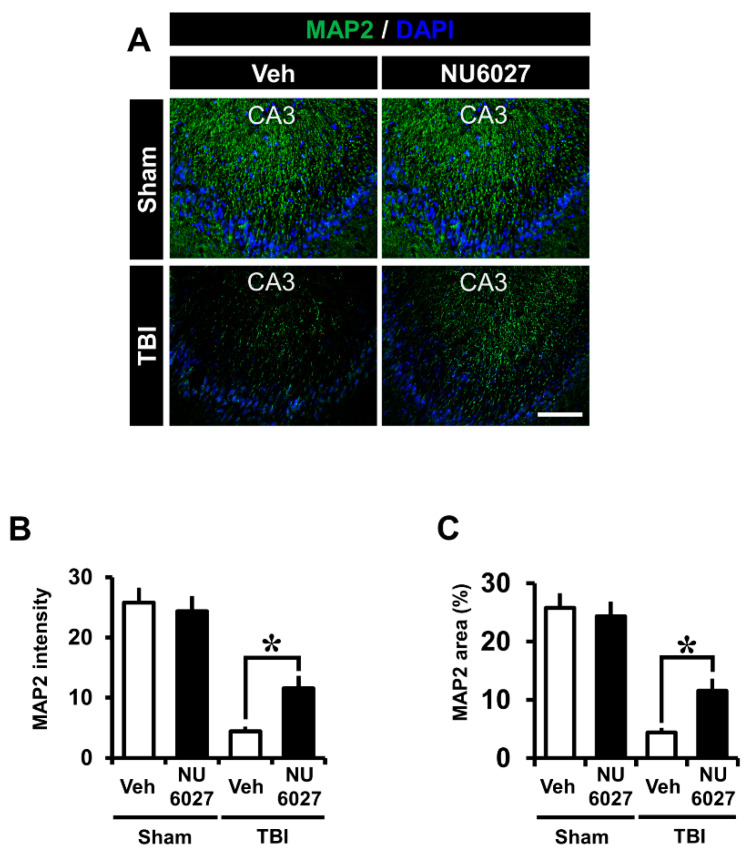
(**A**) Fluorescent images microtubule displays were detected by microtubule-associated protein 2 (MAP2, green color) staining from the hippocampal CA3 regions 24 h after TBI. NU6027 post-treatment decreased the intensity of microtubule loss in the hippocampal CA3 region when compared to the vehicle-treated group. (**B**) The intensity of dendritic loss in the hippocampal CA3 region. The fluorescence intensity displayed a significant difference between groups. (**C**) The bar graph shows the dendritic area percentage in the hippocampal CA3 region (sham-vehicle, *n* = 5; sham-NU6027, *n* = 5; TBI-vehicle, *n* = 6; TBI-NU6027, *n* = 6). Data are mean ± S.E.M. * Significantly different from the vehicle-treated group, *p* < 0.05.

**Figure 3 ijms-21-08256-f003:**
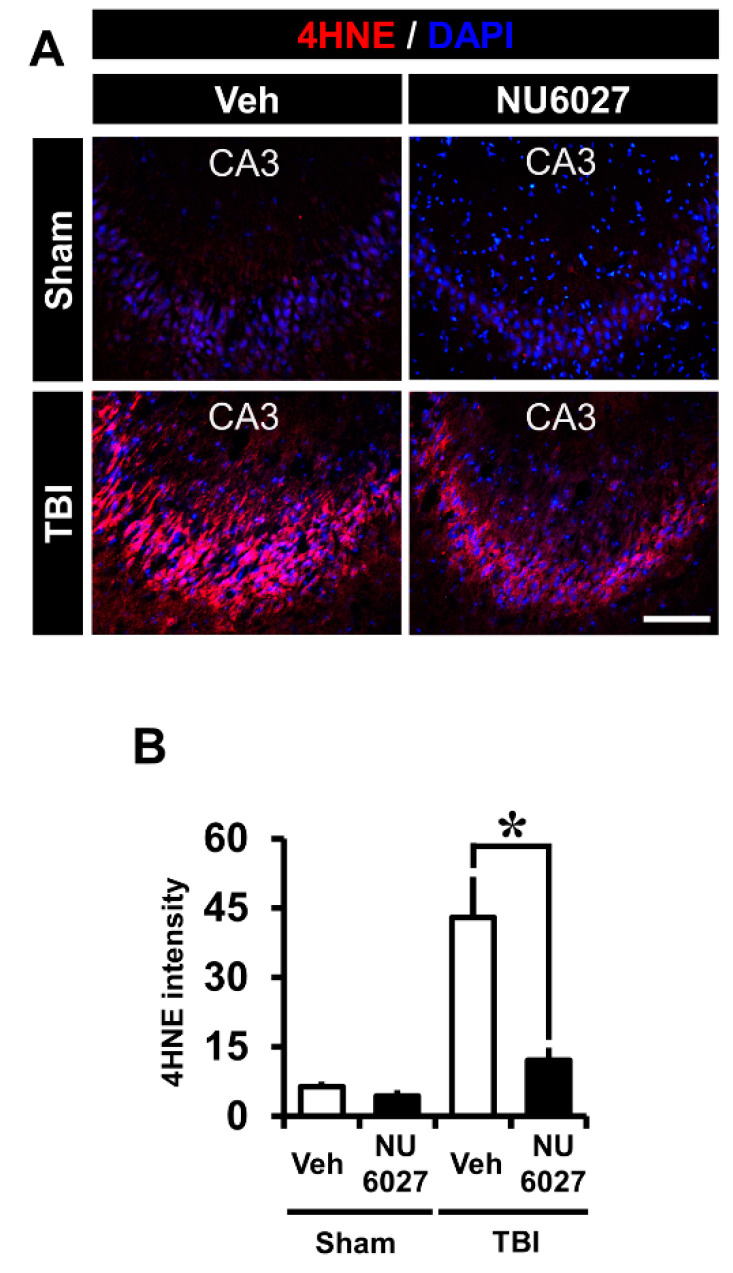
(**A**) Fluorescent images show that oxidative damage was detected by 4-hydroxyl-2-nonenal (4HNE, red color) staining from the hippocampal CA3 region 24 h after TBI. NU6027 post-treatment decreased the intensity of oxidative injury in the hippocampal CA3 region when compared to the vehicle-treated group. (**B**) Bar graph shows the intensity of oxidative injury in the hippocampal CA3 regions. The fluorescence intensity displayed a significant difference between groups (sham-vehicle, *n* = 5; sham-NU6027, *n* = 5; TBI-vehicle, *n* = 6; TBI-NU6027, *n* = 6). Data are mean ± S.E.M. * Significantly different from the vehicle-treated group, *p* < 0.05. Scale bar = 100 μm.

**Figure 4 ijms-21-08256-f004:**
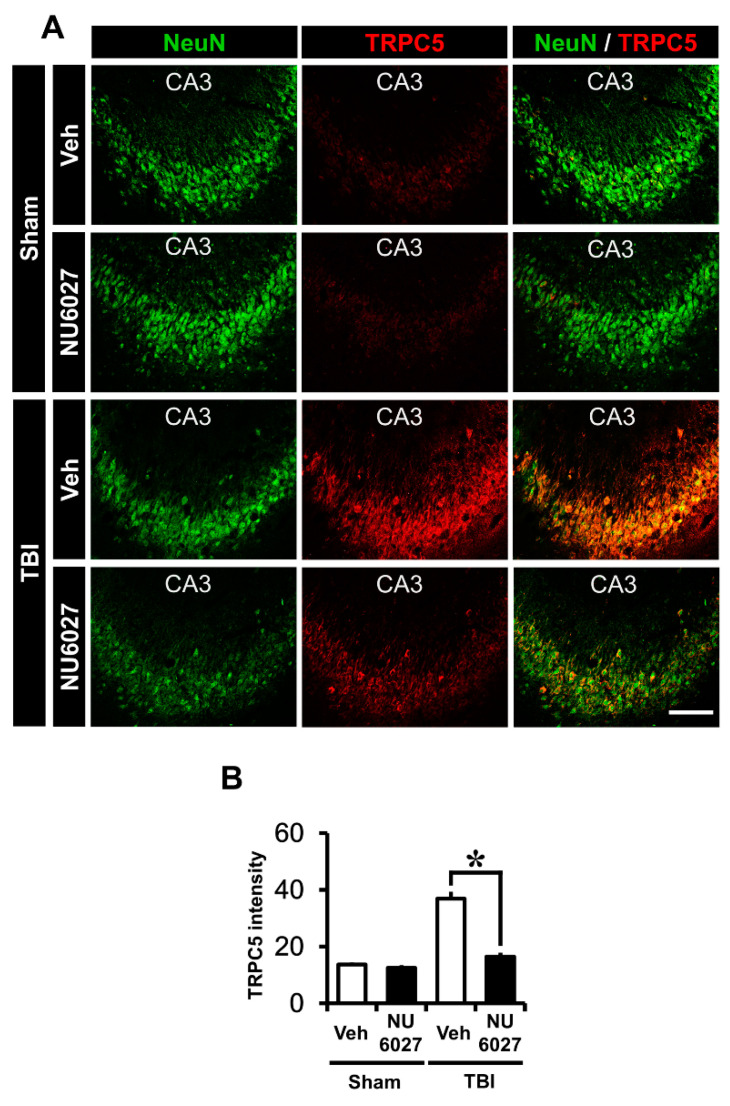
(**A**) Fluorescent images show that the TRPC5 channel expression level was detected by TRPC5 (TRPC5, red color) and counter-staining with neuronal nuclei (NeuN, green color) from the hippocampal CA3 region 24 h after TBI. NU6027 (1 mg/kg) administration after TBI at 24 h decreased a channel expression in the hippocampal CA3 region via inhibition of the TRPC5 channel. (**B**) The intensity of TRPC5 channel expression in the hippocampal CA3 region. The fluorescence intensity displayed a significant difference between groups (sham-vehicle, *n* = 5; sham-NU6027, *n* = 5; TBI-vehicle, *n* = 6; TBI-NU6027, *n* = 6). Data are mean ± S.E.M. * Significantly different from the vehicle-treated group, *p* < 0.05. Scale bar = 100 μm.

**Figure 5 ijms-21-08256-f005:**
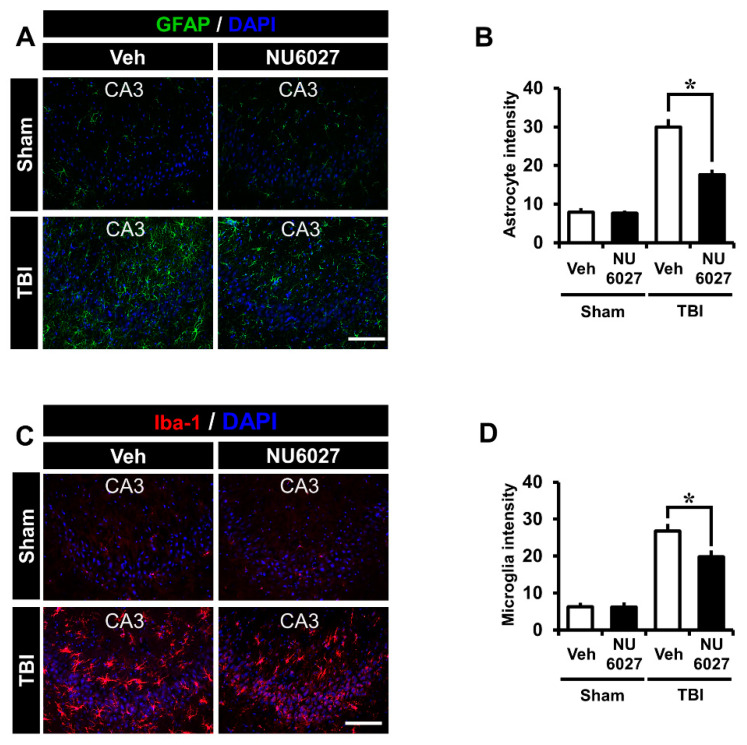
(**A**) Fluorescent images showing that astrocyte activation was detected by GFAP staining from the hippocampal CA3 regions 24 h after TBI. NU6027 (1 mg/kg) post-treatment decreased the intensity of astrocyte activation in the hippocampal CA3 region when compared to the vehicle-treated group (TBI vehicle: *n* = 6; TBI NU6027: *n* = 6). (**B**) The intensity of astrocyte activation in the hippocampal CA3 region. The fluorescence intensity displayed a significant difference between groups. (**C**) Fluorescent images show that microglial activation was detected by Iba-1 staining from the hippocampal CA3 region 24 h after TBI. NU6027 (1 mg/kg) post-treatment decreased the intensity of microglia activation in the hippocampal CA3 region when compared to the vehicle-treated group. (**D**) The intensity of microglia activation in the hippocampal CA3 region. The fluorescence intensity displayed a significant difference between groups (TBI vehicle: *n* = 6; TBI NU6027: *n* = 6). Data are mean ± S.E.M. * Significantly different from the vehicle-treated group, *p* < 0.05. Scale bar = 100 μm.

**Figure 6 ijms-21-08256-f006:**
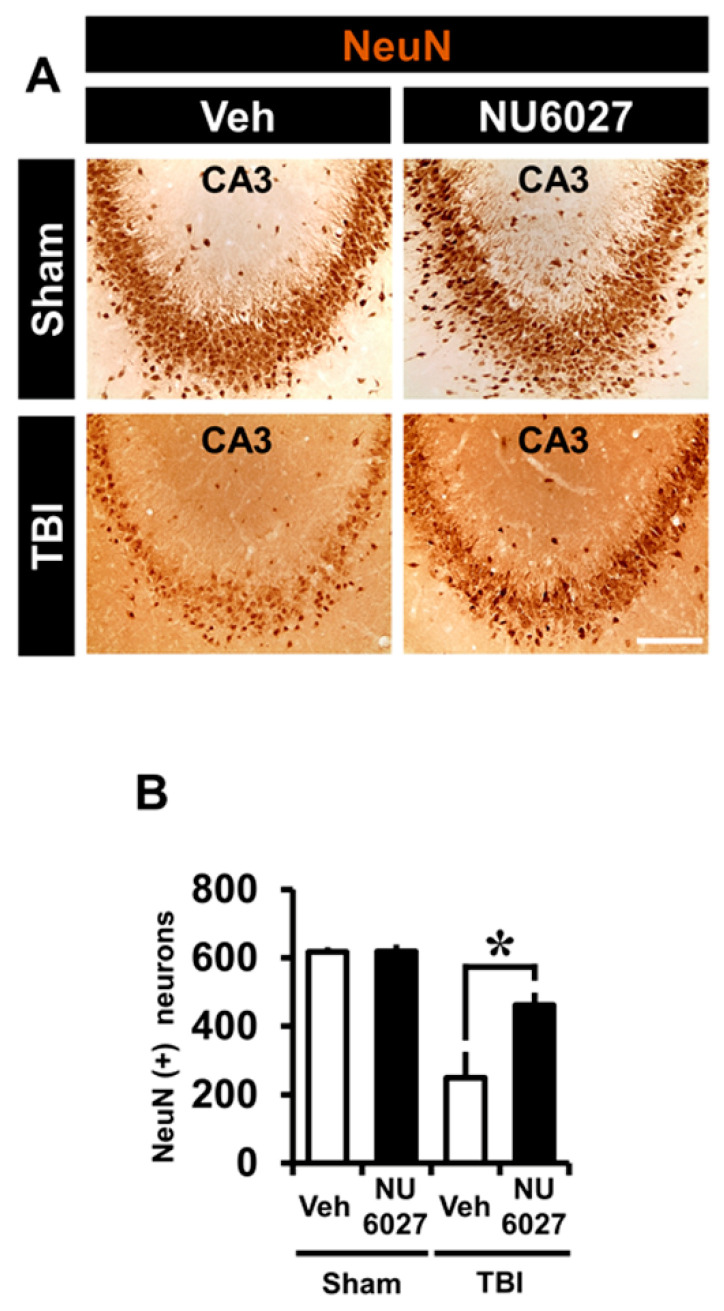
(**A**) Representative images display live neurons detected by NeuN from the hippocampal CA3 region 1 week after TBI. NU6027 (1 mg/kg) post-treatment improved the live neurons in the hippocampal CA3 region when compared to the vehicle-treated group. (**B**) The number of live neurons in the hippocampal CA3 regions (sham-vehicle, *n* = 5; sham-NU6027, *n* = 5; TBI-vehicle, *n* = 5; TBI-NU6027, *n* = 5). Data are mean ± S.E.M. * Significantly different from the vehicle-treated group, *p* < 0.05.

**Figure 7 ijms-21-08256-f007:**
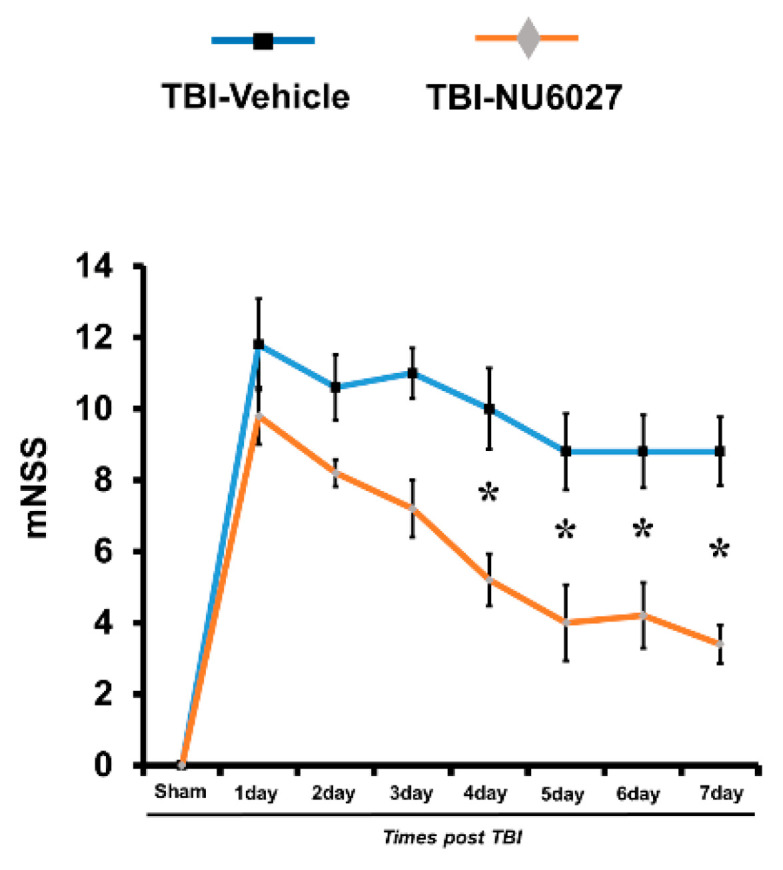
The mNSS grade 18 means all tasks were failed, and grade 0 mean that all tasks have been performed successfully. Determined in rats on consecutive days after TBI (sham-vehicle, *n* = 5; sham-NU6027, *n* = 5; TBI-vehicle, *n* = 5; TBI-NU6027, *n* = 5). Data are mean ± S.E.M. * Significantly different from the vehicle-treated group, *p* < 0.05.

**Figure 8 ijms-21-08256-f008:**
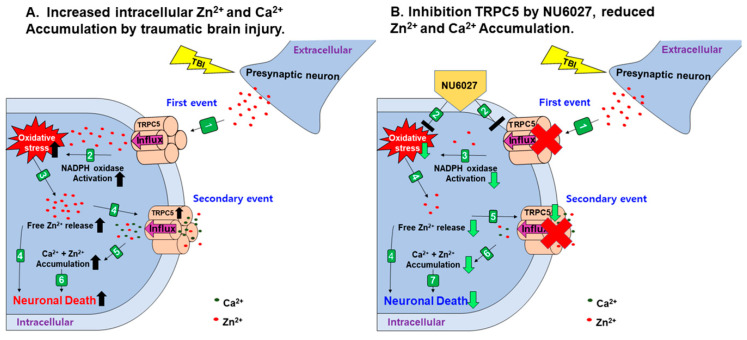
When TBI occurs, edema is caused by a primary inflammatory reaction and the secondary injury oxidative stress occurs, which results in the activation of astrocyte and microglia, loss of many microtubules, and the accumulation of zinc in the neuron. (**A**) (1) Vesicular zinc is released from the presynaptic neuron by TBI [15], and the released vesicular zinc enters the neuron through the TRPC5 channel [14]; (2) NADPH oxidase is activated by vesicular zinc entering the neuron [37,38]; (3) oxidative stress is increased by NADPH oxidase activated by vesicular zinc [37,39,40,41] and excessive free zinc is released into the neuron due to oxidative stress [19,42,43]; (4) TRPC5 channels are activated by the accumulated free zinc [14] and the excess free zinc released causes injury to neurons [15]; (5) zinc and calcium influx due to the activated TRPC5 channel [14,44]; (6) neuronal death occurs. (**B**) (1) Vesicular zinc is released from the presynaptic neuron by TBI [15]; (2) NU6027 is injected immediately after TBI; (3) NU6027 prevents the entrance of vesicular zinc from the TRPC5 channel, thus reducing NADPH oxidase activation and NU6027 also reduces the H_2_O_2_, reducing oxidative stress [14]; (4) oxidative stress is reduced by NU6027, thereby reducing the accumulation of free zinc [14]; (5) the TRPC5 channel is inhibited by the inhibitor NU6027 and there is a reduction of zinc accumulation in neurons [14]; (6) inhibition of the TRPC5 channel reduces the influx of calcium and zinc; (7) neuron death is limited.

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
