# Peer review of "Effects of Transient Receptor Potential Cation 5 (TRPC5) Inhibitor, NU6027, on Hippocampal Neuronal Death after Traumatic Brain Injury"

_ijms, 2020, doi:10.3390/ijms21218256_

Round 1

Reviewer 1 Report

The authors present the results of 2 studies of the effects of a drug, NU6027, on TBI-related tissue damage and motor function. In one cohort of adult male rats, they present evidence that CCI-induced increases in neuronal death, zinc accumulation, dendritic loss, oxidative damage, TRPC5 expression and gliosis 24h after injury are all reduced by post-impact injection of NU6027. In a second cohort, they show that motor function, as assessed by modified neurological severity score (mNSS) 1 week after CCI, was improved in NU6027 treated rats. The study design and conclusions are all appropriate and well-supported. Only minor adjustments are noted below.

  1. The fact that only male rats were used should be highlighted in the main results text and discussed in the discussion.
  2. The drug delivery needs more clarification in the methods. Was there one injection after TBI for the 24h group? How many injections did the 1 week group get?
  3. The authors don’t show in this manuscript that the NU6027 works by blocking zinc influx via non-selective cation channels so any statements to that effect either need to be softened or contextualized as relying on citation to previous literature (e.g. Discussion first sentence; pg 10 lines 288-290). On a related note, more explicit discussion of how specific (or not specific) NU6027 is as a TRPC5 inhibitor is would be beneficial for the reader. Are other TRP channels affected by this drug? What does the direct CDK inhibition effect do?

Author Response

Dear Dr. Luna Liu, Assistant Editor, International Journal of Molecular Sciences,

I appreciate the opportunity to revise this manuscript. I respect the reviewers’ helpful comments and have responded to reviewers’ comments point-by-point, revised as indicated below <yellow green highlights>. I hope this revised manuscript is acceptable for publication in your journal.

Reviewers' Comments to Author:

Reviewer 1

The authors present the results of 2 studies of the effects of a drug, NU6027, on TBI-related tissue damage and motor function. In one cohort of adult male rats, they present evidence that CCI-induced increases in neuronal death, zinc accumulation, dendritic loss, oxidative damage, TRPC5 expression and gliosis 24h after injury are all reduced by post-impact injection of NU6027. In a second cohort, they show that motor function, as assessed by modified neurological severity score (mNSS) 1 week after CCI, was improved in NU6027 treated rats. The study design and conclusions are all appropriate and well-supported. Only minor adjustments are noted below.

  1. The fact that only male rats were used should be highlighted in the main results text and discussed in the discussion. 
  2.  
  3. <Response: We appreciate helpful reviewer’s comments. In this experiment, we used only male rats to minimize the effects of hormones from the female rat’s menstrual cycle. We added this to the revised manuscript.>
  4.  
  5.  
  6. The drug delivery needs more clarification in the methods. Was there one injection after TBI for the 24h group? How many injections did the 1 week group get?
  7.  
  8. <Response: We appreciate reviewer’s comments. In the 24h group, NU6027 was injected into intraperitoneal space once immediately after TBI induction. In the 1-week group, NU6027 was injected immediately after TBI and then once per day for 1 week. We added these details in the revised manuscript.>
  9.  
  10.  
  11. The authors don’t show in this manuscript that the NU6027 works by blocking zinc influx via non-selective cation channels so any statements to that effect either need to be softened or contextualized as relying on citation to previous literature (e.g. Discussion first sentence; pg 10 lines 288-290).
  12.  
  13. <Response: We appreciate this reviewer’s helpful comments. We added following paragraphs at the start of the Discussion section. “In the present study, we tested whether NU6027 inhibits TBI-induced neuronal death via inhibition of transient receptor potential cation channel 5 (TRPC5) channels. TRPC channels subdivided into seven isotypes (TRPC1–TRPC7) and, among them, TRPC5 are abundantly expressed in the rat brain [1-3]. Several studies have demonstrated that TRPC5 over-activation is involve in seizure-induced neuronal death[1, 4, 5]. In the present study, in one cohort of adult male rats, we present evidence that CCI-induced increases in neuronal death, zinc accumulation, dendritic loss, oxidative damage, TRPC5 expression and gliosis 24h after injury are all reduced by post-impact injection of NU6027. In a second cohort, we show that motor function, as assessed by modified neurological severity score (mNSS) 1 week after CCI, was improved in NU6027 treated rats. Therefore, NU6027 may be a potential therapeutic agent for preventing TBI-induced neuronal death.>”
  14.  
  15. On a related note, more explicit discussion of how specific (or not specific) NU6027 is as a TRPC5 inhibitor would be beneficial for the reader. Are other TRP channels affected by this drug?
  16.  
  17. <Response: NU6027 known as specific inhibitor for TRPC5 [4].However, pharmacokinetic specificity for other TRPC subtypes should be tested greater detail in a future study[4]. So far, two studies have demonstrated that calcium and zinc influx is reduced by TRPC5 channel inhibitor NU6027 [4]. Our study has also confirmed that NU6027 reduces zinc accumulation, which is demonstrated by TSQ staining (Figure 1, C).>
  18.  
  19. What does the direct CDK inhibition effect do?
  20.  
  21. <Response: Although NU6027 is a known as a cyclin-dependent kinase 2 (CDK2) inhibitor, TRPC5 targeting inhibition activity has also been verified [4]. The results of several previous studies suggest that CDK inhibition may not be the underlying mechanism for the present study’s neuroprotective effects, since other more potent CDK inhibitors produced no comparable effects published that the protective effect of NU6027 seems to be CDK independent since several other CDK inhibitors were not effective in preventing neuronal death induced by H2O2 [4].>
  22.  
  23.  
  24. Hong, C.; Seo, H.; Kwak, M.; Jeon, J.; Jang, J.; Jeong, E. M.; Myeong, J.; Hwang, Y. J.; Ha, K.; Kang, M. J.; Lee, K. P.; Yi, E. C.; Kim, I. G.; Jeon, J. H.; Ryu, H.; So, I., Increased TRPC5 glutathionylation contributes to striatal neuron loss in Huntington's disease. Brain 2015, 138, (Pt 10), 3030-47.
  25. He, Z.; Jia, C.; Feng, S.; Zhou, K.; Tai, Y.; Bai, X.; Wang, Y., TRPC5 channel is the mediator of neurotrophin-3 in regulating dendritic growth via CaMKIIalpha in rat hippocampal neurons. J Neurosci 2012, 32, (27), 9383-95.
  26. Riccio, A.; Li, Y.; Moon, J.; Kim, K. S.; Smith, K. S.; Rudolph, U.; Gapon, S.; Yao, G. L.; Tsvetkov, E.; Rodig, S. J.; Van't Veer, A.; Meloni, E. G.; Carlezon, W. A., Jr.; Bolshakov, V. Y.; Clapham, D. E., Essential role for TRPC5 in amygdala function and fear-related behavior. Cell 2009, 137, (4), 761-72.
  27. Park, S. E.; Song, J. H.; Hong, C.; Kim, D. E.; Sul, J. W.; Kim, T. Y.; Seo, B. R.; So, I.; Kim, S. Y.; Bae, D. J.; Park, M. H.; Lim, H. M.; Baek, I. J.; Riccio, A.; Lee, J. Y.; Shim, W. H.; Park, B.; Koh, J. Y.; Hwang, J. J., Contribution of Zinc-Dependent Delayed Calcium Influx via TRPC5 in Oxidative Neuronal Death and its Prevention by Novel TRPC Antagonist. Mol Neurobiol 2019, 56, (4), 2822-2835.
  28. Phelan, K. D.; Shwe, U. T.; Abramowitz, J.; Wu, H.; Rhee, S. W.; Howell, M. D.; Gottschall, P. E.; Freichel, M.; Flockerzi, V.; Birnbaumer, L.; Zheng, F., Canonical transient receptor channel 5 (TRPC5) and TRPC1/4 contribute to seizure and excitotoxicity by distinct cellular mechanisms. Mol Pharmacol 2013, 83, (2), 429-38.

Reviewer 2 Report

Comments to the authors

In their manuscript, the authors describe the blockade of TRPC5, which is considered as a zinc/calcium ion channel, by a chemical inhibitor NU6027 suppressed TBI-induced neuronal death in the hippocampus and cognitive impairment of adult rats. The authors suggest the potential application of NU6027 as a therapeutic drug to protect neurons after TBI.

Overall, this study is well-designed and the manuscript is well-documented. Their data are clearly presented for demonstrating neuroprotective effects of NU6027 in this TBI model. This paper is recommended for publication in this journal after some minor revisions, as listed below.

Minor points

  1. Page 9 "Figure8" as the different font
  2. Page 10 Lane 277   "H202" → H2O2
  3. Page 12 Lane 386 "MAP-2" or "MAP2"? as the abbreviation

Author Response

Dear Dr. Luna Liu, Assistant Editor, International Journal of Molecular Sciences,

I appreciate the opportunity to revise this manuscript. I respect the reviewers’ helpful comments and have responded to reviewers’ comments point-by-point, revised as indicated below <yellow green highlights>. I hope this revised manuscript is acceptable for publication in your journal.

Reviewers' Comments to Author:

Reviewer 2

In their manuscript, the authors describe the blockade of TRPC5, which is considered as a zinc/calcium ion channel, by a chemical inhibitor NU6027 suppressed TBI-induced neuronal death in the hippocampus and cognitive impairment of adult rats. The authors suggest the potential application of NU6027 as a therapeutic drug to protect neurons after TBI.

Overall, this study is well-designed, and the manuscript is well-documented. Their data are clearly presented for demonstrating neuroprotective effects of NU6027 in this TBI model. This paper is recommended for publication in this journal after some minor revisions, as listed below.

Minor points

1.Page 9 "Figure8" as the different font

<Response: We appreciate reviewer’s comments. We revised the font in Figure 8.>

2.Page 10 Lane 277   "H202" → H2O2

<Response: We appreciate reviewer’s comments. We corrected it in the revised manuscript.>

3.Page 12 Lane 386 "MAP-2" or "MAP2"? as the abbreviation

<Response: We appreciate the reviewer’s comments. We have corrected to read MAP2 across the manuscript.>
